# Route of Sensitization to Peanut Influences Immune Cell Recruitment at Various Mucosal Sites in Mouse: An Integrative Analysis

**DOI:** 10.3390/nu14040790

**Published:** 2022-02-14

**Authors:** Mélanie Briard, Marine Guinot, Marta Grauso, Blanche Guillon, Stéphane Hazebrouck, Hervé Bernard, Grégory Bouchaud, Marie-Laure Michel, Karine Adel-Patient

**Affiliations:** 1Département Médicaments et Technologies pour la Santé (DMTS), Université Paris-Saclay, CEA, INRAe, Service de Pharmacologie et d’Immunoanalyse, 91190 Gif-sur-Yvette, France; melanie.briard@cea.fr (M.B.); marine.guinot@cea.fr (M.G.); marta.grausoculetto@cea.fr (M.G.); blanche.guillon@cea.fr (B.G.); stephane.hazebrouck@cea.fr (S.H.); herve.bernard-inra@cea.fr (H.B.); 2INRAe, UR1268 Biopolymères, Intéractions et Assemblages (BIA), 44316 Nantes, France; gregory.bouchaud@inrae.fr; 3Micalis Institute, INRAE, AgroParisTech, Université Paris-Saclay, 78350 Jouy-en-Josas, France; marie-laure.michel@inrae.fr

**Keywords:** food allergy, peanut, routes of exposure, BALB/c mice, cellular immune response

## Abstract

Symptom occurrence at the first ingestion suggests that food allergy may result from earlier sensitization via non-oral routes. We aimed to characterize the cellular populations recruited at various mucosal and immune sites after experimental sensitization though different routes. BALB/cJ mice were exposed to a major allergenic food (peanut) mixed with cholera toxin via the intra-gastric (i.g.), respiratory, cutaneous, or intra-peritoneal (i.p.) route. We assessed sensitization and elicitation of the allergic reaction and frequencies of T cells, innate lymphoid cells (ILC), and inflammatory and dendritic cells (DC) in broncho-alveolar lavages (BAL), lungs, skin, intestine, and various lymph nodes. All cellular data were analyzed through non-supervised and supervised uni/multivariate analysis. All exposure routes, except cutaneous, induced sensitization, but intestinal allergy was induced only in i.g.- and i.p.-exposed mice. Multivariate analysis of all cellular constituents did not discriminate i.g. from control mice. Conversely, respiratory-sensitized mice constituted a distinct cluster, characterized by high local inflammation and immune cells recruitment. Those mice also evidenced changes in ILC frequencies at distant site (intestine). Despite absence of sensitization, cutaneous-exposed mice evidenced comparable changes, albeit less intense. Our study highlights that the initial route of sensitization to a food allergen influences the nature of the immune responses at various mucosal sites. Interconnections of mucosal immune systems may participate in the complexity of clinical manifestations as well as in the atopic march.

## 1. Introduction

Food allergy (FA) is an inappropriate immune reaction to food proteins that mainly involves production of specific IgE (type E immunoglobulins) [1]. Production of IgE corresponds to allergic sensitization [1] and the most likely sensitizing route to food antigens is the gastrointestinal tract. After ingestion, food proteins are loaded by antigen-presenting cells spread throughout the intestinal mucosa that migrate then to draining lymph nodes to present the food antigens to naïve T cells. The “normal response” is the induction of regulatory T cells and of immune tolerance. However, in genetically predisposed (i.e., atopic) individuals, and due to specific (micro)environmental signals, naïve CD4^+^ T lymphocytes can differentiate into Type 2 helper cells (Th2) that produce cytokines such as IL-4/5/13 and that will in turn induce production of specific IgE by B lymphocytes. These IgE circulate throughout the body and bind to mucosal mast cells [2]. An allergic reaction can then occur upon further ingestion of the sensitizing food, through activation of these sensitized mast cells.

However, other routes of sensitization to food allergens have been reported, which could explain the occurrence of symptoms after the first known ingestion of the offending food in some patients [3]. Studies in the past decade have brought to evidence that inflamed skin such as that encountered in atopic dermatitis (AD) may be a prominent route for sensitization to some foods. Indeed, food proteins present in the household environment, such as peanut proteins, can penetrate through a damaged or inflamed skin barrier where they are taken up by inflammatory Langerhans cells, leading to Th2 responses and IgE production by B cells [4]. Sensitization may also occur through the airways, especially during food processing (steam) [5]. Immune interconnections between different mucosal sites may then explain that allergic symptoms occur at distant sites. These interconnections may also participate to the “atopic march”, the progression of atopic disorders from AD in infancy to food allergy, allergic rhinitis, and then asthma as children grow up [6,7,8]. Indeed, children suffering from AD have an increased risk of suffering from other atopic diseases and about 35% of them will develop an IgE-mediated food allergy later in life [7].

Another aspect to consider is that structural and functional differences have been described between nasopharyngeal-, skin-, or gut-associated lymphoid tissues [9]. The route of allergen sensitization may therefore influence the nature of the immune responses, i.e., the pattern of produced antibodies and of activated T cells. However, it remains unclear how entering through one route or another influences subsequent allergic reactions, locally or at more distant mucosal sites. The complexity of the mechanisms that underlie allergic inflammatory responses further limits our understanding of the adverse effects that occur in individuals who suffer from allergies.

In a mouse model of experimental sensitization to peanut proteins using cholera toxin as a Th2 adjuvant, we then aimed to further analyze the impact of the route of exposure (gastrointestinal, respiratory, cutaneous, or intraperitoneal) on the cellular actors recruited at various mucosal and immune sites and on food allergy elicitation.

## 2. Materials and Methods

### 2.1. Allergens

For sensitization, a peanut protein extract (PE) was prepared from roasted peanuts [10]. Peanuts were ground with an Ultra-Turrax^®^ Tube Drive (T25 basic, IKA^®^, Werke, Staufen, Germany) and proteins were then extracted in borate buffer (50 mM, pH 9.0, 18 h at 4 °C). After centrifugation (1000× *g* for 15 min) and filtration of the supernatant, PE was obtained. For the oral food challenge (OFC), a protein extract from raw peanut was obtained after defatting of grounded peanuts using acetone and ether, and then a 24 h extraction in a small volume of carbonate buffer (20 mM, pH 9.2).

PE and OFC solutions were characterized by assessment of total protein concentrations (Pierce™ BCA protein assay kit, Thermo Scientific™, Waltham, MA, USA, following the manufacturer’s instructions) and by electrophoresis to ensure the presence of the major peanut allergens [10,11].

### 2.2. Mice

Three-week-old female BALB/cJ mice were purchased from CERJ (Centre d’Elevage René Janvier, Le Genest-Saint-Isle, France), and were housed in filtered cages under normal specific pathogen-free husbandry conditions, with autoclaved bedding and sterile water for 10 days (acclimation). The mice received a diet deprived of peanuts. At the age of five weeks, the mice were individually identified using Radio-frequency identification (RFID) microchips (Biolog-animal) and randomly allocated to experimental groups. All animal experiments were performed according to European Community rules of animal care, and with specific ethical approval from French Minister (authorization number 21846).

### 2.3. Sensitization

In all the groups, exposures were performed once a week for six weeks using PE and 10 µg of cholera toxin (CT; Sigma, Saint-Louis, MO, USA) per administration as a Th2 adjuvant [10,12].

The exposures were performed through either the intragastric (i.g.), intraperitoneal (i.p.), cutaneous (cut.), or respiratory (resp.) route (*n* = 5–8 mice/route of exposure). A group of eight control mice was kept naïve (ctl.) (Figure 1). For i.g., 5 mg of peanut protein, in 200 μL of Phosphate-buffered saline (PBS) buffer, was administered using an animal feeding needle (Popper & Sons, New Hyde park, NY, USA) adapted to the age and weight of the animals [12]. Intraperitoneal exposure (i.p.) was performed by injecting 10 μg of PE in 200 μL of PBS buffer. For cut. exposure, mice were first anesthetized with a mix of ketamine (100 mg/kg; Imalgène^®^ 500, Merial, Lyon, France) and xylazine (10 mg/kg; Rompun^®^ 2%, Bayer Pharma, Puteaux, France) and kept on a heating mat. Then 100 µg of PE diluted in 200 µL of Dimethyl sulfoxide (DMSO) was spread on all sides of the ears. After 30–40 min, the skin was gently cleaned with water to avoid oral contact through grooming. Respiratory exposure was performed by applying 50 μg of PE in 180 µL of PBS, administered by successive deposit of small droplets on the nostril of ketamine-/xylazine-anesthetized mice to avoid swallowing.

### 2.4. Oral Food Challenge (OFC)

One week after the sixth exposure, all the mice including the control group underwent an i.g. challenge with 12 mg of PE to trigger a local (intestinal) allergic reaction.

### 2.5. Single Cell Preparations from Collected Organs

One hour after the OFC, mice were anesthetized and blood and broncho-alveolar lavages (BAL) were collected [13]. Mice were then sacrificed and spleen, lungs, intestine, mesenteric lymph nodes (MLN), mediastinal lymph nodes (MedLN), and ears were collected from each mouse. Organs were immediately placed in RPMI (Sigma-Aldrich, Saint-Louis, MO, USA) and kept on ice until preparation of cellular suspension as detailed in the Appendix A. All samples were treated individually.

### 2.6. mMCP1 Assay

The mouse mast cell protease-1 (mMCP1) concentration was determined in plasma using a commercial kit (mMCP-1 Mouse ELISA Kit, Invitrogen^®^, Waltham, MA, USA), according to the supplier’s recommendations. Each sample was assayed at two dilutions (1/25 and 1/250). Optical density was measured using a spectrophotometer (Epoch^®^, BioTek, Winooski, VT, USA, λ = 414 nm), and analyses were performed using Gen5 software (Gen5.1.09 Software, BioTek, Winooski, VT, USA, https://www.biotek.com/products/software-robotics-software/gen5-microplate-reader-and-imager-software/, accessed on 6 June 2011) [14].

### 2.7. Measurement of PE-Specific IgE and IgG1 in Plasma

Specific IgE and IgG1 were determined as previously described [10,12,14,15] on PE-coated plates. To avoid IgG interaction in IgE measurement, we also performed a “reverse” ELISA assay. Capture antibody (rat anti-mouse IgE, clone LOME-3-Biotech Bio-Rad; 1.5 µg/mL) diluted in 50 mM phosphate buffer pH 7.4 was coated on microtiter plates. After saturation and incubation of the samples, specific IgE were revealed using acetylcholinesterase-labelled peanut proteins [16]. A standard curve was performed on each plate using a pool of plasma from hyper-immunized mice, and concentrations are provided as arbitrary units.

### 2.8. Local and Circulating Cytokine Analysis

Cytokine (IFNγ, IL-5, IL-6, IL-10, IL-13 and IL-17) concentrations were assayed in BAL and plasma using xMAP^®^ Luminex Technology’s kits and apparatus from Bio-Rad (BioPlex™ Cytokine Assay, Bio-Rad; Hercules, CA, USA, Bioplex™200) and following the manufacturer’s recommendations.

### 2.9. Cellular Population Analysis

Cell counts and viability were determined using a NovoCyte Flow Cytometer (ACEA Biosciences, San Diego, CA, USA) and 7-aminoactinomycin D (7-AAD, Interchim, Montluçon, France). Immunophenotyping of T cells (including homing receptors), dendritic cells (DC), innate lymphoid cells (ILC), and inflammatory cells was performed using four pre-optimized antibody panels presented in the Appendix A. Data acquisition was performed on an Attune™ NxT Flow Cytometer (Thermo Scientific™, Waltham, MA, USA) or on a NovoCyte Flow Cytometer, depending on the cell subtypes, and analyses were performed using FlowJo^®^ (Version 10, ACEA Biosciences, San Diego, CA, USA). The gating strategies are illustrated in Appendix A.

The various analyses described above were carried out on all the samples at the same time.

### 2.10. Statistical Analysis

Food allergy induction was first confirmed by comparing concentrations of specific IgE and IgG1, cytokines, and mMCP1 in plasma from exposed versus naïve mice (non-parametric Kruskal–Wallis and Dunn’s post-test). Then, we aggregated all the data from the cellular analysis to perform multivariate analysis. We first performed a descriptive analysis (principal component analysis, PCA) of all data obtained from each individual. This offered an overview of the variables and individuals, to identify potential outliers (none identified), and to assess the variables that were the most explicative of the whole dataset. Individuals with some missing data were not considered in PCA (three i.g., three i.p., and three ctl. mice). Corresponding missing data resulted from the quantity of cells not always being sufficient in some tissues to carry out relevant analysis in flow cytometry analysis. Non-supervised clustering was also tested (agglomerative hierarchical clustering, AHC): AHC gathers the closest individuals by considering all the variables. Homogeneity of the repartition of the different individuals in the different clusters was tested to preliminarily identify if some routes of exposure led to a clustering of corresponding individuals. Actually, if a route of exposure did not influence the type of immune response induced, the individuals would be shared equivalently within the different clusters. Then, we modelled the data using supervised partial least square discriminant analysis (PLS-DA), with the exposure route as the explanatory variable. If a model was constructed, it meant it is possible to distinguish the individuals depending on the route of exposure based on the analyzed variables. Discriminant variables (i.e., the ones on which the “route of exposure” effect relied) were then identified with model-calculated “variable important in projection” (VIP) value (VIP +/−SD > 1).

Univariate analyses were performed in parallel. The non-parametric Kruskal–Wallis test first identified immune constituents evidencing differences within experimental groups (*p* < 0.05). Paired analysis using Dunn’s test was then performed to compare all groups to the control group (* *p* < 0.05; ** *p* < 0.01; *** *p* < 0.001).

All analyses were performed using XLSTAT^®^ software (version 2020.3, Addinsoft, Paris, France) and a graph drawn with GraphPad Prism 9.0.0. Univariate plots are shown as box-and-whisker plots (median, upper and lower quartile).

## 3. Results

### 3.1. Assessment of Food Allergy Induction

We first assessed the sensitization level induced in the different groups of mice, by comparing the concentrations of peanut-specific IgE and IgG1 antibodies and Th1/Th2/Th17 (i.e., IFNγ/IL-5/IL-17) and inflammatory/regulatory cytokines (i.e., IL-6/IL-10) in plasma.

Compared to the control mice, peanut-specific IgE antibodies were significantly induced in mice exposed through the i.g., respiratory, or i.p. route, but not in mice exposed through the cutaneous route (Figure 2A). Peanut-specific IgG1 antibodies were higher in mice exposed through the i.p. or the respiratory route, compared to control mice (Figure 2A). No increased concentrations of cytokines was evidenced in plasma from the exposed groups compared to naïve mice (not shown).

A significant intestinal allergic reaction was induced by the OFC in i.g.- and i.p.-sensitized mice, as evidenced by significantly increased concentrations of mMCP1 in plasma from the corresponding mice (Figure 2B). Despite efficient sensitization, no elicitation of the allergic reaction was observed after the OFC in respiratory-exposed mice.

Therefore, the sensitization route influenced the intensity of sensitization and, at equivalent level of sensitization, the susceptibility to develop a clinical allergic reaction after an intra-gastric challenge. Then, to go deeper in the characterization of the induced immune response and to identify its specificity depending on the sensitization route, we performed multivariate analysis of the cellular immune components assessed in the different tissues and lymph nodes collected from all these mice.

### 3.2. Analysis of Immune Cellular Components in Tissues, Lymph Nodes, and BAL Fluids

#### 3.2.1. Non-Supervised Analysis

DC, T cells (including their homing receptors), and ILC subtypes, as well as inflammatory cells (i.e., neutrophils, eosinophils, and macrophages) were analyzed by flow cytometry in the different organs and lymph nodes and in BAL collected at sacrifice from all mice. In total, 93 immune cellular constituents were first analyzed through non-supervised multivariate analyses, i.e., PCA and then AHC.

The first two dimensions of the PCA represented 41.84% of the total variance of the dataset. The first dimension of the PCA (24.24% of total variance,) tended to separate i.p.- and resp.-exposed mice from the other groups, and these last mice were clearly distinguished from the others with the first and second dimension (17.60% of the total dataset variance) (Figure 3A). Many cellular data obtained in the lungs, such as CD8^+^, CD4^+,^ and ILC cells, were the most contributing variables of the first dimension (Appendix A). The variables that mainly contributed to the second dimension were CD4^+^ cells and inflammatory cells in the lungs, ears, and BAL.

In line with these observations, cut.-, i.p.-, and resp.-exposed mice were classified in separate and well-defined clusters in non-supervised AHC clustering (not shown, chi-square *p* value = 5.04 × 10^−7^).

#### 3.2.2. Multivariate and Univariate Supervised Analysis

##### Data Modelling and Identification of Discriminant and Significant Variables

PLS-DA modelling with immune cellular data was then performed, with the exposure route as the explanatory variable. A two-component model with low predictive values (R²Xcum = 0.251, R²Ycum = 0.403) was obtained. Mice exposed through cutaneous, respiratory, and i.p. routes were correctly classified. Conversely, i.g.-exposed mice and control mice were misclassified, mostly within cut. or i.p. groups (Figure 3B). VIP values obtained for all the cellular parameters are provided in Appendix A. Cellular data obtained in BAL (e.g., CD4^+^, CD8^+^, and inflammatory cells) and lungs (ILC and DC) were the more discriminating constituents in the modelling (VIP +/− SD > 1).

As the high inflammatory profile in the respiratory tract of mice exposed through the respiratory route may have masked some subtle changes in the other groups during our multivariate analysis, we performed the multivariate statistical analyses without the data from the resp. group. However, results were comparable to those previously obtained (R²Xcum = 0.219, R²Ycum = 0.517), i.e., mice exposed though the i.p. and cutaneous routes were well separated whereas control and i.g.-exposed mice were overlapping.

In parallel, pairwise univariate analysis were performed to identify cell constituents significantly different between groups, which were further tested using pairwise analysis comparing each group to the ctl. group (naïve mice), with correction for multiple testing.

Univariate graphs of the most discriminant (high VIP) and significant (*p*-value < 0.05 between groups and ctl.) parameters are shown to visualize the differences. Corresponding data are shown and discussed below. No difference was evidenced in the ear tissue and in spleen, whatever the group (not shown).

##### Analysis in Intestinal Tissue and Associated Lymph Nodes

On the one hand, no significant difference was evidenced between exposed and control mice for frequencies of DC subtypes or inflammatory cells in MLN, lamina propria (LP), or within iEL (data not shown). A significant increased frequency of CCR2^+^ cells within CD4^+^ and CD8^+^ cells was observed in the LP from i.p.-exposed mice compared to control mice (Appendix A). The number of ILC in LP was also significantly increased in mice exposed through the i.p. route (Figure 4A). This resulted in a significant increase of frequency of both ILC2 and ILC3 (notably ILC3 CCR6^+^ NKp46^−^, Figure 4B) in the LP from mice exposed through the i.p. route compared to control mice (Figure 4B), and was associated with a decrease of ILC2 frequency within iEL (Appendix A). Few modifications were observed for i.g.-exposed mice, except for an increased frequency of NK (NKp46^+^ NK1.1^+^) cells within iEL (Appendix A).

Modifications of the immune system in intestinal tissue were also noticed in the resp. group, but the changes were not exactly the same as the ones observed in the i.p. group—the CD4^+^ and CD8^+^ T populations were not altered, nor those of ILC2, whereas ILC3 and NK cell frequencies were also increased in the LP from mice exposed through the respiratory route compared to control mice (Figure 4B,C). Interestingly, ILC count and frequencies of ILC3 and CCR6^+^ NKp46^−^ cells were also increased in the LP (Figure 4A,B), and frequency of NK cells was increased within iEL (Appendix A), in mice exposed through the cutaneous route, despite the absence of IgE/IgG1 induction.

##### Analysis in the Respiratory Tract (BAL and Lung Tissue) and Associated Lymph Nodes (MedLN)

In the lung tissue, we observed an increased frequency of CD11b^+^CD103^−^ DC subtype and of pDC (SiglecH^+^) in mice exposed through the respiratory route, and to a lesser extent in mice exposed through the cutaneous route (Figure 5A). Total ILC counts and ILC1 frequency in the lung tissue were significantly lower in i.p., i.g., and resp. groups than in control mice (Figure 5B).

In the BAL, we observed a high increase of CD45^+^ cells frequency in mice exposed through the respiratory route compared to control mice (Figure 6), which resulted from an influx of various inflammatory cells, i.e., eosinophils, neutrophils, and macrophages (Figure 6, not significant for eosinophils). This influx was accompanied by an influx of both CD4^+^ and CD8^+^ T cells that showed significant increased expression of various homing receptors such as CCR2, CCR4, CCR7, and CCR8 (Figure 7), and of activated T cells (CD4^+^CD25^+^) (Figure 8). The local inflammation evidenced in mice exposed through the respiratory route was associated with a significant increase of IL-17 concentration in BAL (Figure 9), with no increase for other cytokines assayed (not shown).

On the other hand, a less intense, albeit significant, influx of CD4^+^ T cells expressing CCR2, CCR4, CCR7, and CCR8 (Figure 7A,B) and of CD4^+^ CD25^+^ T cells (Figure 8) was also observed in BAL from mice exposed through the cutaneous route compared to control mice.

No significant difference was evidenced between the groups and the control mice for DC cell frequencies in MedLN (Appendix A).

## 4. Discussion

Observation in humans has evidenced that sensitization to some food allergens may occur via the skin or the respiratory mucosa [4,17], leading to symptoms after the first ingestion of the offending food. This suggests inter-connection between the various mucosal surfaces, which may also participate in the natural history of allergy such as the atopic march. Various studies in animals also evidenced that the route of administration affects the nature of the induced immune responses [18,19,20]. Food allergy can be induced through various routes of sensitization in mouse models, but no comprehensive analysis has been performed to deeply analyze the association between exposure routes and induced cellular responses. Using mice and peanut as a model food allergen, we therefore aimed to analyze the immune cellular components recruited at various sites following exposure through different routes. We used the Th2 adjuvant cholera toxin for all exposure routes in order to focus on exposure route independently of the adjuvant effect. Mice were exposed through the i.g., respiratory, cutaneous, or i.p. route, and levels of sensitization and of elicitation of the allergic reaction after a food challenge were first analyzed. In parallel, a comprehensive immunophenotyping at different mucosal sites was performed and analyzed through multivariate and univariate approaches.

First, we evidenced that peanut sensitization is efficiently induced after exposure through the i.p., i.g., or respiratory routes, as shown by high specific IgE and IgG1 antibody concentrations in plasma. A significant allergic reaction occurred after an OFC in i.p. and i.g. mice, but not in the resp. group despite the systemic sensitization. Conversely, no sensitization nor allergic reaction was elicited in mice exposed through the cutaneous route. This was not in line with previous study [21]. Of note, here we did not alter the epithelium to induce a skin inflammation such as by tape stripping [20]. Moreover, despite the absence of sensitization in periphery, we evidenced that mice exposed through the cutaneous route displayed an increase of some cell populations at distant sites, i.e., in the respiratory tract (i.e., CD11b^+^CD103^−^ DC in the lung and CD4^+^ in the BAL) and to a lesser extent in the intestine (i.e., ILC in LP). These results thus suggest interconnections between the skin and the intestinal mucosa, in line with previous studies in both mice and humans [19,20], but also between skin and lungs.

Inversely, i.g.-exposed mice evidenced allergic sensitization and allergic reaction upon food challenge, but few modifications were observed at any studied mucosal sites, even at the sensitizing site where only NK cells within iEL were increased. This suggests not only little mobility of cells from the gut-associated lymphoid tissue to other mucosa, but also a limited local inflammatory/adaptive response except for IgE-sensitized mast cells (not directly assessed in the present analysis albeit suggested through mMCP1 release upon oral challenge). Conversely, mobility of Tregs induced by gavages in the intestine to the lung has been evidenced [22].

On the contrary, in addition to a systemic sensitization, mice exposed through the respiratory tract evidenced a high inflammatory profile at the sensitization site. This was evidenced by significant influx of CD11b^+^CD103^−^ DC and pDC SiglecH^+^ cells in the lungs, and of inflammatory cells (e.g., neutrophils, eosinophils, and macrophages) and CD4^+^ and CD8^+^ cells in the BAL. Quite surprisingly, these T cells expressed the homing receptors CCR2, CCR4, CCR7, and CCR8, which promoted the migration of cells to different sites, respectively, to the broncho-alveolar area, the skin, the secondary lymphatic nodes, and the gut. Although we did not evidence such T cells in the LP, respiratory-exposed mice showed increased ILC and NK cells in the intestine, thus suggesting a lung to intestine connection.

The local inflammation of mice exposed through the respiratory route was associated with high concentrations of IL-17 in BAL, in line with recent studies evidencing that IL-17 play an important role on driving allergic inflammation [23,24] and can contribute to the induction of Th2 cells and eosinophil accumulation, in addition to IgE production [24]. Surprisingly, no elicitation of the allergic reaction following an oral challenge was observed in the respiratory-exposed mice. This absence of allergic reaction upon OFC may have resulted from a low density of IgE-bearing mast cells in the intestine of respiratory-exposed mice, which was not assessed in the present study, although we observed increased frequency of innate lymphoid cells in the intestine and an efficient systemic sensitization. This further underlines the importance of the local immune components present at the elicitation site and may explain the disconnection of sensitization and elicitation of the allergic reaction observed in some patients and in some animal models [25]. In our respiratory-exposed mice, a respiratory challenge would probably have amplified the already existing local inflammation.

## 5. Conclusions

In conclusion, our results evidenced a connection between the different mucosal sites for various cellular components, independently of the systemic sensitization. These connections depend on the initial route of exposure, and may involve different cellular components, of both the immune and adaptive immune systems. Recruitment of these cellular components may render mucosal sites distant from the sensitization site ready for elicitation and inflammation after further encounters with the sensitizing food allergen. It may also render those sites prone to sensitization to other allergens, thus participating in the atopic march. Such model and associated integrative analyses may then help in understanding the events and actors involved in allergic sensitization to food allergens and in the atopic march [26,27,28]. Further studies with other food allergens are underway to analyze if the observed responses are allergen-dependent.

## Figures and Tables

**Figure 1 nutrients-14-00790-f001:**
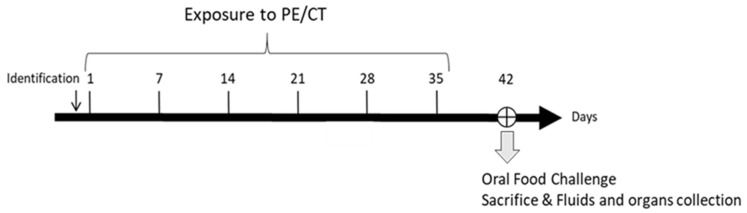
Experimental schedule. Mice were exposed once a week for 6 weeks to a peanut protein extract (PE) mixed with cholera toxin (CT) through intra-gastric (*n* = 8), intra-peritoneal (*n* = 8), cutaneous (*n* = 7), or respiratory (*n* = 5) routes. An oral food challenge was performed on day 42 and then fluids and organs/tissues were collected.

**Figure 2 nutrients-14-00790-f002:**
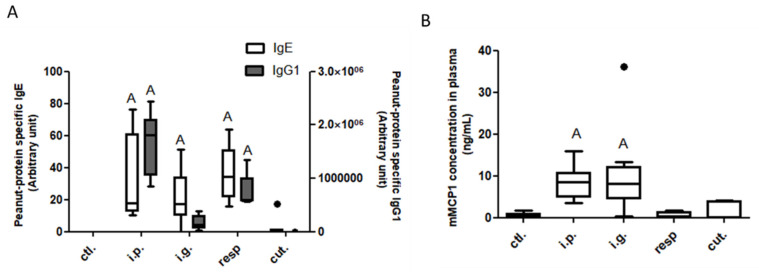
Assessment of food allergy induction after exposure through different routes: Peanut-specific IgE (open bars) and IgG1 (gray bars) (**A**) and mMCP1 (**B**) concentrations in plasma collected 1 h after an oral food challenge from mice previously exposed through the intraperitoneal (i.p.), intragastric (i.g.), respiratory (resp.), or cutaneous (cut.) routes, and in non-exposed mice (ctl.; control mice). “A”: indicates a significant difference between indicated group and control group (*p* < 0.05 using non-parametric Kruskal–Wallis and Dunn’s post-test). Bold black dots represent outliers.

**Figure 3 nutrients-14-00790-f003:**
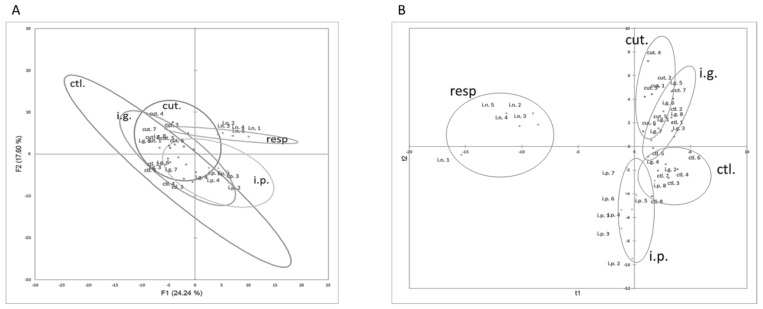
Multivariate analysis of immune cellular constituents assessed for each mouse exposed through the i.p., i.g., resp., or cut. routes. (**A**) Graph of individuals on the first two dimensions of non-supervised PCA. (**B**) Graph of individuals after modelling of all the data using supervised PLS-DA and route of exposure as the explicative variable.

**Figure 4 nutrients-14-00790-f004:**
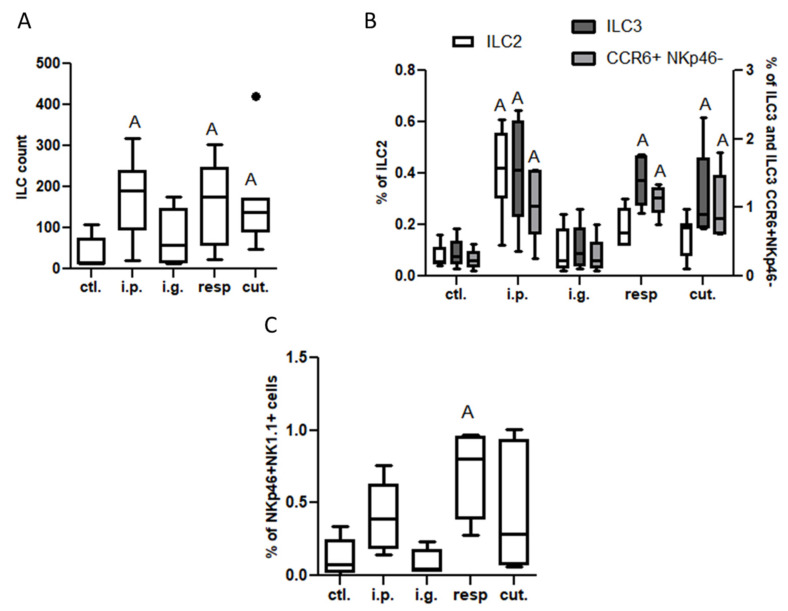
Innate lymphoid cells in lamina propria from mice exposed through the i.p., i.g., resp., or cut. routes compared to ctl. mice: (**A**) ILC count (CD45^+^Lin^−^CD4^−^CD127^+^ cells); (**B**) ILC2 (ST2^+^NK1.1^−^ cells within ILC, empty bars), ILC3 (ST2^−^NK1.1^−^ cells selected within ILC; dark gray bars), and CCR6^+^NKp46^−^ ILC3 (light gray bars) frequencies; and (**C**) NK cells (NK1.1^+^NKp46^+^ cells selected within CD45^+^lin^−^, then CD4^−^CD127^−^ cells). All frequencies are expressed using CD45^+^ cells as a reference. “A” indicates a significant difference between indicated group and control mice (*p* < 0.05 using non-parametric Kruskal–Wallis and Dunn’ post-Test). The bold black dot represents an outlier.

**Figure 5 nutrients-14-00790-f005:**
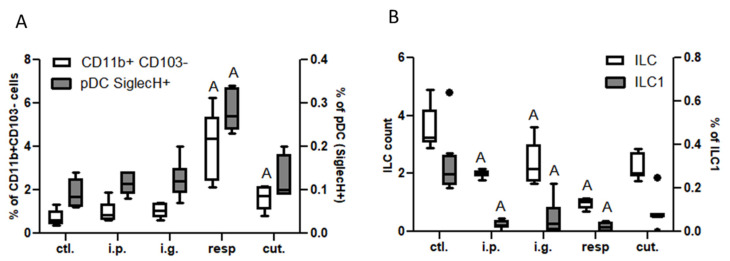
Dendritic cells and ILC in lung tissue: (**A**) CD11b^+^CD103^−^ DC subtypes (empty bars) and pDC (gray bars) in the different groups of exposed mice. DC were defined as CD45^+^CMHII^+^CD11c^+^ cells and DC subtypes were defined based on CD11b and CD103 expression. pDC were defined as SiglecH^+^ cells within CD11b^−^CD103^−^ cells. (**B**) ILC counts (CD45^+^Lin^−^CD4^−^CD127^+^ cells, empty bars) and ILC1 frequencies (ST2^−^NK1.1^+^ cells within ILC, gray bars) in the different group of mice. All frequencies are expressed using CD45^+^ cells as a reference. “A” indicates a significant difference with ctl. group (*p* < 0.05 using non-parametric Kruskal–Wallis and Dunn’s post-test). Bold black dots represent outliers.

**Figure 6 nutrients-14-00790-f006:**
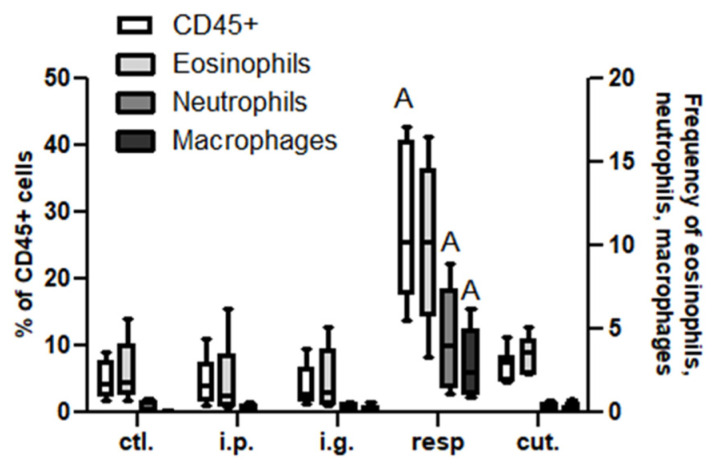
Inflammatory cells in BAL: Frequencies of CD45+ (empty bars), eosinophils (light gray bars), neutrophils (dark gray bars), and macrophages (black bars) in BAL from different groups of exposed mice. Inflammatory cells were defined within CD45^+^CD3^−^ cells as neutrophils (Ly6C^+^Ly6G^+^), eosinophils (SiglecF^+^CCR3^+^ within LyG6^−^), and macrophages (Ly6C^−^F4/80^+^ within LyG6^−^). All frequencies are expressed using CD45^+^ cells as a reference. “A” indicates a significant difference between indicated group and control group (*p* < 0.05 using non-parametric Kruskal–Wallis and Dunn’s post-Test).

**Figure 7 nutrients-14-00790-f007:**
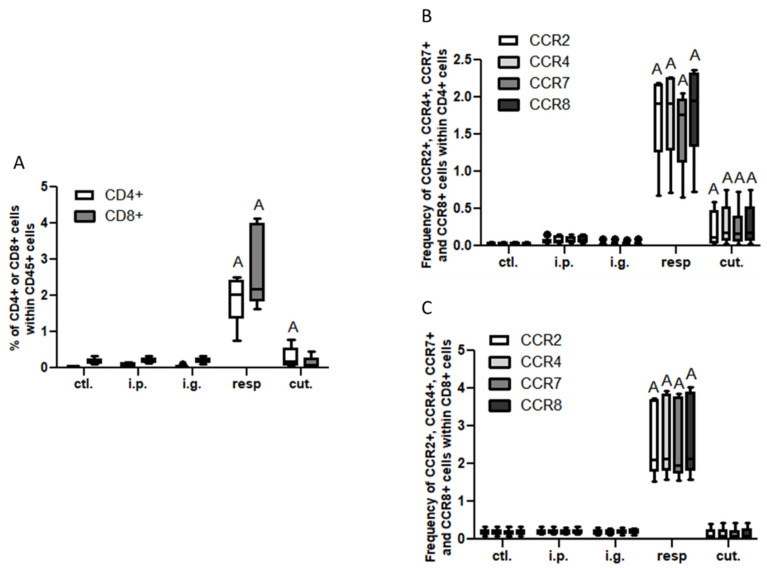
T cell influx in BAL: (**A**) Frequencies of T lymphocytes CD4^+^ (CD8^−^, empty bars) and CD8^+^ (CD4^−^, gray bars). Frequency of CCR2^+^ (empty bars), CCR4^+^ (light gray bars), CCR7^+^ (dark gray bars), and CCR8^+^ (black bars) within CD4^+^ (**B**) or within CD8^+^ (**C**) in BAL from mice exposed through the different routes. CD4^+^ and CD8^+^ cells were selected within CD45^+^CD3^+^ gated cells. “A” indicates a significant difference between indicated group and control group (*p* < 0.05 using non-parametric Kruskal–Wallis and Dunn’s post-Test).

**Figure 8 nutrients-14-00790-f008:**
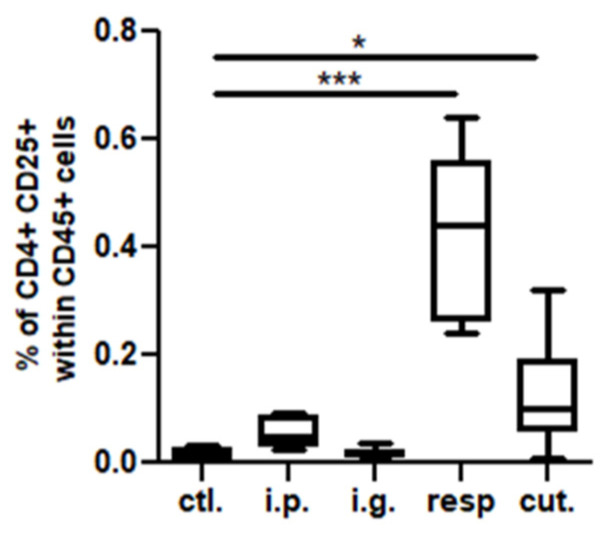
Activated T cells in BAL: CD4^+^CD25^+^ cell concentrations in BAL from mice exposed through the different routes. * *p* < 0.05; *** *p* < 0.001: Using non-parametric Kruskal–Wallis test and Dunn’s post-test when comparing all groups to control one (non-exposed mice).

**Figure 9 nutrients-14-00790-f009:**
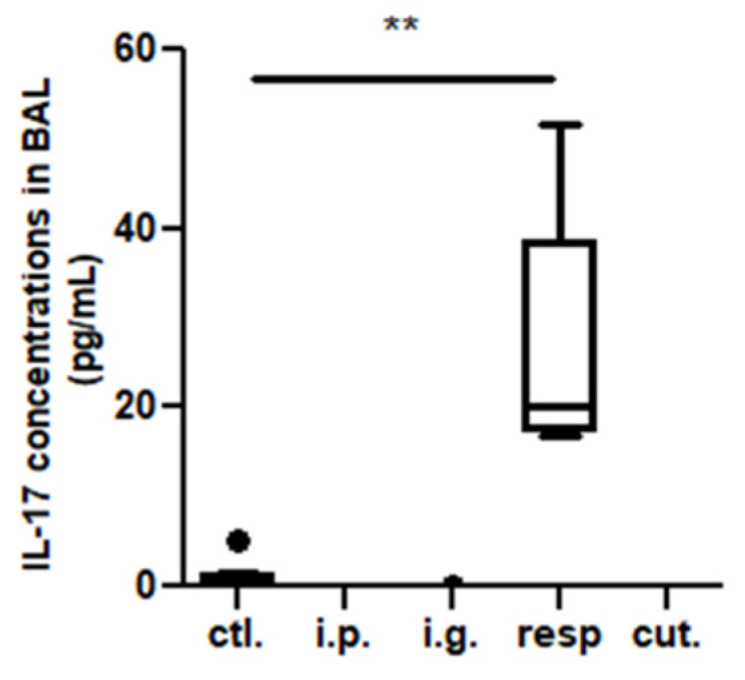
IL-17 concentrations in BAL from mice exposed through the different routes. ** *p* < 0.01: Using non-parametric Kruskal–Wallis test and Dunn’s post-test when comparing all groups to control one (non-exposed mice). The bold black dot represents an outlier.

## Data Availability

The raw data supporting the conclusions of this manuscript are available from the corresponding author to any qualified researcher on reasonable request.

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
