# Peer review of "Route of Sensitization to Peanut Influences Immune Cell Recruitment at Various Mucosal Sites in Mouse: An Integrative Analysis"

_nutrients, 2022, doi:10.3390/nu14040790_

Round 1

Reviewer 1 Report

The manuscript is of high interest to the scientific community.
The strategies and methodologies presented are widely used by the authors as evidenced by the number of self-citations in the bibliography.
The writing is clear and the results are well presented. However, the manuscript would be improved with the inclusion of figures S8 and S9 in the main body of the manuscript, as well as the inclusion of some of the results (not shown) referred to by the authors, namely: Page 9 line 325 to 328 "The local inflammation evidenced in mice exposed through the respiratory route was associated with a significant increase of IL-17 concentration in BAL (Sup-plementary figure 9), with no change for other cytokines assayed (not shown)" and page 10, line 348 to 349 " No significant difference was evidenced between groups and control mice for DC or inflammatory cells frequencies in MedLN (not shown). " 

Reviewer 2 Report

Manuscript ID: nutrients-1571132
Type of manuscript: Article
Title: Route of sensitization to peanut influences immune cells recruitment at various mucosal sites in mouse: an integrative 3 analysis.

This is a well-written original article focused on the role of the route of sensitization (intra-gastric, respiratory, cutaneous or intra-peritoneal) to a food allergen that may influence the nature of the immune responses at different mucosal sites. The authors also used an interesting approach and multiple statistical methods to analyze the data.

I have only minor comments for the authors.

  • Methods 2.3, and 2.4. did control group underwent an i.g. challenge with peanut proteins (OFC)? It is not clear.
  • Often throughout the text, and especially in the Results section, the message "Error! Reference source not found". Probably it is a comment of the editorial office,

However please complete the references to the figures, tables that are referred to in the text, because it much easier to follow the results.

  • Line 241, shouldn’t be “that”not than?
  • Line 403, “Surprisingly, no symptom following an oral challenge was….”What symptom did you mean?
  • Lines 367-372, Please elaborate more on the possible reasons for successful respiratory sensitization but lack of allergy after OFC.
